# Flow Virometry Quantification of Host Proteins on the Surface of HIV-1 Pseudovirus Particles

**DOI:** 10.3390/v12111296

**Published:** 2020-11-12

**Authors:** Jonathan Burnie, Vera A. Tang, Joshua A. Welsh, Arvin T. Persaud, Laxshaginee Thaya, Jennifer C. Jones, Christina Guzzo

**Affiliations:** 1Department of Biological Sciences, University of Toronto Scarborough, 1265 Military Trail, Toronto, ON M1C 1A4, Canada; jonathan.burnie@mail.utoronto.ca (J.B.); arvin.persaud@mail.utoronto.ca (A.T.P.); laxsha.thaya@mail.utoronto.ca (L.T.); 2Department of Cell and Systems Biology, University of Toronto, 25 Harbord Street, Toronto, ON M5S 3G5, Canada; 3Department of Biochemistry, Microbiology, and Immunology, Faculty of Medicine, University of Ottawa, Flow Cytometry and Virometry Core Facility, Ottawa, ON K1H 8M5, Canada; vtang@uottawa.ca; 4Translational Nanobiology Section, Laboratory of Pathology, Center for Cancer Research, National Cancer Institute, National Institutes of Health, Bethesda, MD 20892, USA; joshua.welsh@nih.gov (J.A.W.); jennifer.jones2@nih.gov (J.C.J.)

**Keywords:** human immunodeficiency virus (HIV), flow cytometry, calibrated flow virometry, molecules of equivalent soluble fluorophore (MESF), virion-incorporated proteins, integrin α4β7, CD14, CD162 (PSGL-1), CD81 (tetraspanin)

## Abstract

The HIV-1 glycoprotein spike (gp120) is typically the first viral antigen that cells encounter before initiating immune responses, and is often the sole target in vaccine designs. Thus, characterizing the presence of cellular antigens on the surfaces of HIV particles may help identify new antiviral targets or impact targeting of gp120. Despite the importance of characterizing proteins on the virion surface, current techniques available for this purpose do not support high-throughput analysis of viruses, and typically only offer a semi-quantitative assessment of virus-associated proteins. Traditional bulk techniques often assess averages of viral preparations, which may mask subtle but important differences in viral subsets. On the other hand, microscopy techniques, which provide detail on individual virions, are difficult to use in a high-throughput manner and have low levels of sensitivity for antigen detection. Flow cytometry is a technique that traditionally has been used for rapid, high-sensitivity characterization of single cells, with limited use in detecting viruses, since the small size of viral particles hinders their detection. Herein, we report the detection and surface antigen characterization of HIV-1 pseudovirus particles by light scattering and fluorescence with flow cytometry, termed flow virometry for its specific application to viruses. We quantified three cellular proteins (integrin α4β7, CD14, and CD162/PSGL-1) in the viral envelope by directly staining virion-containing cell supernatants without the requirement of additional processing steps to distinguish virus particles or specific virus purification techniques. We also show that two antigens can be simultaneously detected on the surface of individual HIV virions, probing for the tetraspanin marker, CD81, in addition to α4β7, CD14, and CD162/PSGL-1. This study demonstrates new advances in calibrated flow virometry as a tool to provide sensitive, high-throughput characterization of the viral envelope in a more efficient, quantitative manner than previously reported techniques.

## 1. Introduction

Despite decades of research, the human immunodeficiency virus (HIV) remains a prominent global issue, with 38 million people estimated to be living with the virus in 2019 [1]. Targeting the viral envelope glycoprotein, gp120, in vaccine designs has proven to be difficult due to the protein’s extensive glycosylation, frequent mutations, and conformational camouflage of neutralizing epitopes [2,3,4,5,6]. A less commonly discussed feature of the viral envelope that may be of interest when informing new therapeutics or vaccine designs is the myriad of cellular proteins embedded within it. The acquisition of cellular proteins into the HIV envelope through the process of viral egress (budding) is a well-described phenomenon and select studies have described the functional impact of specific incorporated proteins on viral fitness (CD54/ICAM-1, MHC Class II/HLA-DR, LFA-1, and CD59) [7,8,9,10,11,12,13]. While a broad range of cellular proteins have been identified as being present in the HIV envelope [8,13,14,15,16,17], few have been extensively characterized in terms of their absolute quantities on virus particles or impacts on viral fitness. Recently, this field has been stimulated by the identification of two previously undescribed proteins in the viral envelope, which have seemingly opposite effects on infection: integrin alpha 4 beta 7 (α4β7) and P-selectin glycoprotein ligand (PSGL-1/CD162) [18,19,20,21].

Cellular proteins within the viral envelope may be desirable targets for new therapeutic or vaccine strategies, since they can be selectively acquired by virions and have also been suggested to outnumber gp120 on the virus [18,22]. Interestingly, in animal models of HIV infection, treatment of macaques with an anti-α4β7 monoclonal antibody suppressed SIV replication and protected animals in transmission studies [23,24,25], leading us to speculate whether some of the protection afforded by the anti-α4β7 treatment could be from targeting virions with incorporated α4β7. Furthermore, the therapeutic use of monoclonal antibodies targeting host proteins has been proven safe in humans, as a humanized version of the same anti-α4β7 used in animal studies is an FDA-approved treatment for irritable bowel disease (IBD), and when administered to patients with IBD and concomitant HIV, the antibody elicits a reduction in lymphoid aggregates [26]. Thus, virion-incorporated cellular proteins may provide new antiviral targets on the surface of virions that can be targeted with monoclonal antibodies or other directed therapeutics.

Traditional methods used to identify host proteins in the HIV envelope include bead- or plate-based antibody virus capture assays and immunogold labeling paired with transmission electron microscopy (TEM) [15,16,17,27,28,29]. The former can be described as bulk techniques, while the latter provides information on individual virions. While informative, bulk techniques sample the total virus population along with co-isolated extracellular vesicles (EVs) and fail to detect pertinent information regarding small, distinct viral subpopulations. Indeed, it is well-described that over the course of HIV-1 infection, a diverse population of virus quasispecies are generated due to the high rate of virus mutation, selective pressure by host immune responses, and tissue compartmentalization of infection [30,31,32,33,34]. Furthermore, during infection the majority of particles produced are defective [35,36,37,38], with most methods unable to discern these non-infectious particles from the few infectious ones. On the other hand, while immunogold labeling provides information on individual virions, it is time-intensive and is not ideal for acquiring data in a high-throughput manner. In addition, while EM has previously been used to quantify proteins in the HIV envelope [28,39], the harsh fixatives used in sample preparation and the inaccessibility of viral antigens due to virus immobilization on grids may hinder the reliability of this data.

Throughout the last decade, few efforts have been made to use the well-established cellular technique flow cytometry to study proteins on HIV virions. Traditionally, the size of viral particles has made the analysis of viruses with flow cytometry challenging, since viruses typically fall within the range of instrument noise for most cytometers [40]. However, Arakelyan et al. demonstrated that detection of cellular and viral proteins in the HIV envelope on individual virus particles was possible through a technique they coined “flow virometry” [41]. This technique employed capturing virions on magnetic nanoparticles (MNPs) with fluorophore-conjugated antibodies against gp120; captured virus particles were then stained using a second fluorophore-labeled antibody against the antigen of interest on the virus for phenotyping. The subsequent population was detected via fluorescence triggering, since fluorescence detection is more sensitive than light scattering with most traditional cytometers [41]. While this method has opened the door to many new questions about virions that were difficult—if not impossible—to address previously, the caveat of this method is that the detection and analysis of virus particles is only possible after viruses are captured with an MNP-bound antibody. Thus, those virus particles that do not express the antigen detected by MNP-antibody capture will be excluded from downstream analysis, which may lead to bias in the assay. Similarly, binding of 15 nm MNPs to the virus may reduce the ability of other antigens to be successfully stained. Due to these limitations, enhanced methods for single-particle viral characterization by cytometers are needed.

One method that has become more common for detecting viruses with flow virometry is employing fluorescent dyes to label the membrane or nucleic acids of the virus particles [42,43,44,45,46,47,48]. Similar detection strategies with fluorescently tagged fusion proteins have also allowed for the removal of coupling to magnetic beads during analysis [49,50]. Importantly, the ability to detect non-fluorescent virus preparations by light scattering alone has been performed only on highly specialized cytometers, which are more sensitive in their ability to detect scattering, or using giant viruses (i.e., >400 nm diameter) [47,51]. Furthermore, while the detection of fluorescently labeled viruses by flow virometry has been performed in several studies [49,52,53,54,55,56,57], the ability to quantify viral surface proteins reliably and consistently continues to be a major challenge in the field, due to contaminating nanoscale particles (EVs) in virus preparations and instrument variability [58]. Quantitative determination of virus surface proteins remains of paramount importance for HIV-1, in particular since it may help inform which proteins on the surface of a virus are the most attractive targets for novel treatment or vaccine strategies in terms of both protein abundance and unique epitope accessibility during different stages of infection.

Herein, we report a novel protocol utilizing flow virometry to detect HIV-1 pseudoviruses by light scattering and the quantification of three different virion-incorporated host proteins (integrin α4β7, CD14 and CD162 (PSGL-1)) by fluorescence, without the requirement of coupling viruses to magnetic particles, fluorescent dyes, or viral concentration procedures. To our knowledge, this is the first report involving staining the surface of HIV particles for cellular proteins without the need for additional methods to enhance the detection and discrimination of virus populations. In this study, we estimate the total number of virion-incorporated proteins on individual virus particles (reported in calibrated units—molecules of equivalent soluble fluorophore, MESF; molecules of equivalent reference fluorophore, MERF) and demonstrate the stepwise staining process and controls needed to perform these analyses. The goal of this work is to increase the accessibility and use of flow virometry in order to enable other labs to generate comparable data that can collectively advance the field.

## 2. Materials and Methods

### 2.1. Cell Culture and Virus Production

HEK293 cells obtained from the NIH AIDS Reagent Program (ARP) were maintained in complete media comprised of DMEM (Wisent, St-Bruno, Quebec, Cat#319-005-CL), 10% fetal bovine serum (FBS; Wisent, Cat#098150), 100 U/mL penicillin, and 100 µg/mL streptomycin (Life Technologies, Burlington, Ontario, Canada, Cat#15140122), and were grown in a 5% CO_2_ humidified incubator at 37 °C. For flow virometry virus production, cells were seeded at a density of 10^6^ cells/mL in 6 well plates in complete media. To produce control HIV pseudoviruses, cells were transfected (Polyjet SignaGen^®^, Frederick, MD, USA) with 2 μg of SG3^Δenv^ pDNA (NIH ARP) after cells had reached 70% confluence. To produce viruses with host proteins CD14 and CD162 (denoted as CD14+ and CD162+), cells were co-transfected with 1 μg of host protein and 1 μg of SG3^Δenv^ pDNA, while for α4β7+ viruses, 0.5 μg of each integrin subunit (α4 and β7) was co-transfected with 1 µg of the SG3^Δenv^ pDNA. All transfections were performed with a 1:3 ratio of plasmid DNA (μg) to transfection reagent (μL). Plasmids expressing human α4 and β7 integrin subunits, CD14, and CD162 were obtained from OriGene Technologies Inc. (Rockville, MD, USA), Sino Biological (Beijing, China) and Addgene (Watertown, MA, USA), respectively. Six hours after transfection, the media were removed from the wells (to discard any initial viral progeny without incorporated host proteins) and replaced with complete DMEM containing EV-depleted FBS. Since FBS is known to contain a large number of bovine extracellular vesicles that could impact analysis of small particles [59], we ultracentrifuged FBS at 73,000× *g* for 24 h before use to reduce the number of contaminating EVs in our culture media. Viruses were harvested 48 h after transfection, shipped overnight on ice to the University of Ottawa Flow Cytometry and Virometry Core Facility, and stored at 4 °C to be stained and analyzed by flow virometry within 24 h. HEK293 cells were also mock-transfected with 2 μg of an empty vector (Sino Biological, Cat#CV011) and supernatants were collected for use in studies to determine the level of EVs induced upon transfection of HEK293 cells, as we expected EVs could overlap in side scattering profiles with virion-containing supernatants derived from virus transfected cells.

### 2.2. Cellular Flow Cytometry

Flow cytometry used to assess cell surface expression of host proteins was performed using a BD LSRFortessa (San Jose, CA, USA) instrument with FACS Diva software (San Jose, CA, USA), and all data were analyzed using FlowJo software version 10.7.1. (San Jose, CA, USA). HEK293 cells were stained with primary mouse anti-human monoclonal antibodies against α4β7 (clone ACT-1 [60]; NIH ARP), CD14 (clone M5E2; BD Bioscience, Sparks, MD, USA), CD162 (clone KPL-1; BD Bioscience), and CD81 (clone JS-81; BD Bioscience) for 30 min. After primary antibodies were removed by washing, staining with an R-phycoerythrin (PE) conjugated F(ab’)2-goat antimouse IgG secondary antibody which recognizes IgG heavy and light chains (Invitrogen, Carlsbad, CA, USA; Cat#A10543) was performed for 20 min. All antibodies were used at a concentration of 2 μg/mL for cellular staining.

### 2.3. Flow Virometry

Flow virometry was performed using a Beckman Coulter CytoFLEX S (Mississauga, ON, CA) with standard optical configuration. A 50 mW 561 nm laser with 561–585/42 bandpass filter was used for the detection of the fluorophore R-phycoerythrin (PE) and an 80 mW 405 nm laser with 405/10 and 450/45 bandpass filters was used for side-scattered light (SSC) and fluorophore Brilliant Violet^TM^ 421 (BV421) detection, respectively. Gain and threshold optimization for detection of virus and calibration beads was performed as described previously [58]. All virus samples and controls were acquired at a sample flow rate of 10 μL/min for 1 min, with the exception of double-stained virus samples and controls, which were acquired for 2 min. Volumetric calibrations were performed on the instrument using the calibration application in the CytExpert (Mississauga, ON, CA) acquisition software. The virus particle concentrations in cell-free supernatants were estimated based on gated events from serially diluted unstained samples that were collected for 1 min at 10 μL/min. Virus suspensions, collected as cell-free supernatants, were diluted to 10^9^ particles/mL and stained with PE-conjugated monoclonal antibodies against α4β7, CD14, and CD162 or BV421-conjugated mouse anti-human CD81 (same clones as Section 2.2) before being further diluted with PBS (to reduce coincidence) for analysis by FV. For select experiments (as indicated), this staining was performed using a 1 h staining protocol that was described previously [58]. To reduce the background noise and the amount of antibody required for labeling, viruses were stained at 5 × 10^8^ particles/mL with a final concentration of 0.2–0.25 µg/mL of antibody at 4 °C for 22 h (i.e., overnight incubation). Following staining, samples were diluted 1000-fold to give a final concentration of 5 × 10^5^ particles/mL for analysis. BD Quantibrite PE beads (San Diego, CA, USA; Cat# 340495, lot 91367) and Spherotech 8 Peak Rainbow calibration particles (Green Oaks, IL, USA; Cat# RCP-30-5A, lot AF01) were used for fluorescence calibration, while NIST-traceable size standards (Thermo Fisher Scientific; Carlsbad, CA, USA) were used for light scattering calibration. Calibration was performed using FCM_PASS_ software (https://nano.ccr.cancer.gov/fcmpass) as previously described [58,61]. Detailed information on the fluorescent and light scatter calibration can be found in the FCM_PASS_ output report in Appendix A**.** Experiments were conducted in compliance with the MIFlowCyt-EV framework [62], using the MIFlowCyt-EV checklist in Appendix A.

### 2.4. Virion Capture Assay

Immunomagnetic bead-based virion capture was performed as previously described [18,63], with 25 μL of protein G Dynabeads (Life Technologies; Cat# 10004D), which were armed with 0.5 μg of mouse monoclonal antibody (antibodies described above in Cellular Flow Cytometry) for 20 min at room temperature and then washed with 10% FBS–PBS to remove unbound antibodies. At the start of capture assays, the virus input was normalized across all viruses tested, with inputs of equal virus volumes (150 µL) all at the same concentration (35 ng/mL of p24). Viruses were incubated with antibody-armed beads for 1–2 h at room temperature to allow virus capture. Beads were then washed three times with 10% FBS–PBS and once with 0.02% FBS–PBS to extensively remove unbound virus particles. The bead-associated virus was then treated with 0.5% Triton X-100 to lyse the captured virions for p24 quantification by ELISA. Data analysis was performed using Prism v. 8.4.2 (GraphPad, San Diego, CA, USA). The background level of virion capture for each virus type was assessed by virion capture with an isotype control antibody, mIgG (clone MOPC-31C, BD Biosciences). The nominal level of background capture (range of 100–300 pg/mL) was removed from each data point before graphing where indicated.

### 2.5. p24 ELISA

The quantification of HIV p24 was performed in captured virus lysates and virus-containing supernatants with the DuoSet p24 ELISA kit following the manufacturer’s (R&D Systems, Minneapolis, MN, USA) instructions. Absorbance readings were performed on a Synergy NEO 2 multimode plate reader (BioTek, VT, USA) equipped with Gen 5 software (v. 3.08).

### 2.6. Electron Microscopy

Visualization of pseudovirus particles was performed on sections of virus-producing HEK293 cells. Briefly, cells were grown on sterilized 18 mm diameter coverslips cells and were transfected with 0.83 μg of SG3^Δenv^ pDNA (as described above) once cells had reached 70% confluency. Eight hours after transfection, the media were replaced, and 24 h after transfection the media were removed from the coverslips and the cells were washed with PBS. Samples were subsequently fixed with 2.5% glutaraldehyde in 0.1 M PB for 2 h at room temperature followed by an overnight 4 °C incubation before TEM processing. For EM processing, samples were washed 3 times with 0.1 M cacodylate buffer (pH 7.3) before being reduced with 1% OsO4 in cacodylate buffer and stained with 4% uranyl acetate. After staining, samples underwent ethanol dehydration and infiltration in Quetol–Spurr resin before polymerization in fresh resin at 70 °C for 2 days. Samples were then sectioned at 80 nm before staining on copper grids with uranyl acetate and lead citrate. After processing, samples were imaged on a Hitachi H-7500 transmission electron microscope (TEM, Hitachi High-Technologies, Fukuoka, Japan) equipped with a Megaview G2 CCD camera (Olympus, Toronto, ON, CA). Images were acquired from areas of free virions (i.e., virions that were not cell-associated) to validate our pseudovirus particle production methods.

## 3. Results

### 3.1. Validation of Virus Stocks for Flow Virometry

Since our goal was to use flow virometry to detect cellular proteins on the surface of virus particles, we first established a transfection-based virus production model to generate viral particles that were positive or negative for our proteins of interest (Appendix A). We generated HIV pseudovirus particles (termed virus herein for simplicity) using the subtype B, envelope-deficient viral clone SG3^Δenv^ [64]. Three human cellular proteins, namely integrin α4β7, CD14, and CD162, were selected as model cellular proteins to test in this study, since they were previously well-described as being incorporated in the HIV-1 envelope [18,19,20,65,66]. In Figure 1A, we visualized our virus by TEM, which demonstrated that the virions have an electron-dense capsid region and that they are relatively uniform in shape and size (Figure 1A), validating the reliability of our virus production system. Next, cell surface staining and flow cytometry analyses were performed to validate cell surface expression of the transfected proteins on the producer cells from which the viruses were harvested (Figure 1B). This step was critical to ensure that nascent virions were able to incorporate the cellular proteins of interest during the process of budding. From flow cytometry, we observed high levels of cell surface expression for each transfected cellular antigen (as detected by mean fluorescence intensity—MFI). To confirm that the virions incorporated these cellular proteins, we performed immunomagnetic capture assays (Figure 1C), as previously described [18,63]. The level of capture with anti-CD162 antibody was markedly higher than what was seen with anti-CD14 and anti-α4β7 antibodies, despite the levels of surface expression (MFI values) being relatively similar for CD162 and CD14. We expect that this discrepancy may be due to specific cellular mechanisms promoting the incorporation of CD162 into HIV virions [21]. Virion-incorporated α4β7 detected by capture assays was lowest among the three cellular antigens tested. This was expected, since α4β7 also displayed the lowest levels of cell surface expression. Based on these data, we were able to confirm that our virus production system generated pseudovirus particles with or without our proteins of interest, as required for flow virometry analyses in this study.

To validate our ability to detect these viruses on the CytoFLEX cytometer as performed previously [51], three different dilutions of viruses were run through the cytometer in addition to a media control (Figure 1D; Appendix A). A single virus population as defined by SSC was observed, indicating that the viruses are monodisperse and that viral aggregates are largely absent from the sample. With successive two-fold dilutions, the number of viral events (particles/mL) within the common gate was shown to decrease two-fold, as expected. This validated that our gated population is representative of single virus particles, since coincidental events and viral aggregates would be less likely to dilute out reproducibly.

### 3.2. Flow Virometry Data Calibration

After confirming that our virus particles could be consistently detected on the cytometer, we began optimizing the staining of our virus samples and standardizing the output of our flow virometry data. Through an iterative process of sample staining and acquisitions on the cytometer, we determined that our proteins of interest (α4β7, CD14, and CD162) were very sensitive to PFA fixation, and thus the samples needed to be stained before being fixed and acquired on the cytometer. We chose PE-conjugated monoclonal antibodies to stain the cellular proteins of interest, as absolute quantitation of cell surface antigens using PE and commercially available reference beads had been previously described [67,68,69,70]. Since PE is such a large fluorophore, antibody preparations typically contain a 1:1 ratio of PE to antibody [67]; thus, when calibrated to a scale of molecules of equivalent soluble fluorophore (MESF) using commercially available reference beads, one can infer the number of individual antibodies bound to cells or viruses, which can provide an estimate of the number of proteins present [70,71].

To ensure that our flow virometry data could be used for accurate quantitation of virion-incorporated proteins and be displayed in standard units that can be compared across different institutions and instruments, we calibrated our axes using commercially available reference beads and software (Figure 2A–D). Polystyrene and silica beads (NIST-traceable size standards) were used with light scatter modeling software to calibrate to standard units of effective scattering cross-section (Figure 2A,B). The units of scattering cross-section are standard units independent of refractive index and size, however they are dependent on the collection angle, which can be accounted for between instruments if the modeling parameters are reported, as they have been here (Appendix A, Appendix A). BD Quantibrite PE and Sphero 8 Peak Rainbow calibration particles were used as references for the calibration of PE and BV421 arbitrary fluorescence intensity (Figure 2C,D), respectively. Calibration of raw FCS file data to FCS files containing calibrated units was performed using FCM_PASS_ software [61,72]. Quality control plots outputted from the FCM_PASS_ calibration showed a very high correlation between arbitrary and standard units (R^2^ > 0.99) (Appendix A). Assessment of the fit of our data for scatter modeling by comparing the predicted versus acquired SSC values for the different diameter sizing beads run on the cytometer showed that our data fit well with the scatter modeling performed by FCM_PASS_ (Figure 2E). By applying FCM_PASS_ to the FCS files generated throughout this study, PE fluorescence was converted so as to be reported in molecules of equivalent soluble fluorophore (MESF; Figure 2F) values, a standardized method of reporting fluorescence data that allows for analysis and comparison of sample acquisitions, regardless of the instrumentation and parameters (e.g., laser power, collection angles, detector settings, filter configuration, etc.) used [58]. Due to the lack of availability of BV421 MESF reference beads, CD81 was calibrated to a scale of molecules of equivalent cascade blue (MECSB). Due to BV421 and CSB not having identical emission spectra, BV421 fluorescence was expressed in MECSB units of equivalent reference fluorophore (ERF). With fluorescence references for our labeling antibodies and standardized presentation of our flow data established, we next began to optimize our antibody staining parameters.

### 3.3. Optimization of Antibody Labeling and Staining Parameters for Flow Virometry

Since viruses are several orders of magnitude smaller than cells, there are far fewer epitopes available for labeling on the viral surface. Due to this, the use of optimal antibody staining techniques is of utmost importance to ensure that small deviations with labeling and detection, which may not be detectable in cellular flow cytometry, do not lead to large inaccuracies in quantifying viral proteins in flow virometry. Furthermore, while employing multiple wash steps to remove unbound antibodies, dye and debris in cellular flow cytometry are common; since viruses are too small to be pelleted effectively on a conventional centrifuge, wash steps are often omitted in small particle flow cytometry or flow virometry [49,51,58]. This is especially important to consider, since detection of unbound antibody alongside unlabeled virus can lead to coincidental detection and a high level of background fluorescence, which can mask dim biological signals (Appendix A). With this in mind, we performed a titration of a PE-conjugated antibody to determine the amount of background fluorescence generated by the antibody diluted in PBS alone (Appendix A). The anti-α4β7 antibody was chosen for this purpose, since it was conjugated in-house and displayed more unbound PE than the other antibodies used in this study. Additional titrations were then performed for each antibody to maximize specific binding to viruses with incorporated proteins, while ensuring minimal background fluorescence on control viruses (those without incorporated proteins). Representative data for these titrations is shown for the anti-CD14 antibody (Figure 3A). It should be noted that the stained samples in Figure 3 underwent dilutions after labeling (as described in methods), ensuring that the positive fluorescence signal was due to detection of stained virus and not coincidence. Through antibody titration, the optimal concentration for staining that separated our CD14+ HIV particles fully from the background was determined to be 2.5 μg/mL when a 1 h staining incubation protocol at room temperature was used (Figure 3A). The shift in fluorescence caused by anti-CD14 staining was not present when the control virus was stained, indicating the specificity and biological relevance of the labeling.

While being able to detect our host proteins of interest with this protocol was a significant breakthrough in the application of flow virometry to HIV, we noticed a large amount of background fluorescence was present despite antibody titration and a 250-fold dilution of the stained virus (2 × 10^6^ particles/mL) before acquisition on the cytometer (Figure 3B), likely due to coincidence from the high level of unbound fluorescent antibody in the sample (Appendix A). While additional dilutions of the stained virus did allow for some reduction of this background fluorescence (Figure 3B), this was not ideal, since our sample was now very dilute, with a virus particle concentration of about 1.25 × 10^5^ particles/mL at the dilution that returned the background fluorescence to unstained virus levels. Since small particles have relatively few external epitopes available for labeling, increases in background fluorescence may hinder the ability to successfully resolve them, and therefore hinder the quantitation of proteins associated with each single virion. Therefore, in an attempt to further reduce background fluorescence caused by high concentrations of antibody, we extended the antibody staining time to an overnight incubation at 4 °C while using 10-fold less antibody (0.25 μg/mL). This extended staining incubation was tested since overnight antibody incubations are commonly used among a range of antibody assays, such as immunofluorescence microscopy and Western blotting protocols, with the same intent to reduce background fluorescence and maximize labeling. We observed that this staining protocol allowed for similar levels of staining as compared with the 1 h high-concentration (2.5 μg/mL) staining protocol at RT, but most importantly we observed reduced coincidence, as evidenced by a decrease in fluorescence background (Figure 3C, comparing grey boxed data with right side plots). To ensure that we achieved a similar number of virus events despite the reduced background, we doubled the acquisition time on the cytometer (from 60 to 120 s) for samples processed with the overnight labeling technique. With this optimized labeling technique, we proceeded to stain all of our viruses using overnight incubations with 0.2–0.25 μg/mL of antibody at 4 °C for subsequent assays.

### 3.4. Double Labeling of Cellular Antigens on HIV-1 Pseudoviruses

Given that we had established the technical parameters (and relevant controls) for virion staining and analysis, we continued on to perform the biologically relevant work, which was to stain the surface of HIV virions for incorporated host proteins. We stained three different viruses with incorporated host proteins (α4β7+ HIV, CD14+ HIV, and CD162+ HIV) with PE-labelled antibodies against the incorporated proteins. As seen in Figure 4A, virus populations were identified with gates based on homogeneity in SSC. We observed a notable shift in fluorescence for each stained virus population above the background fluorescence (Figure 4A, bottom row). Interestingly, each stained virus population had a different distribution and range of associated fluorescence values. We speculate that these differential population characteristics may be due to biological differences within the virus, whereby the incorporation of certain proteins appears to be more homogeneous on viruses, as observed with CD14, while other proteins seem to be incorporated at variable levels on virions, as observed for α4β7. No positive labeling was detectable for the control HIV samples (Figure 4A, top row), indicating that only the specific host proteins on the surface of the viruses were being effectively labeled. For the CD162+ viruses, two distinct scattering populations were stained with anti-CD162 (Figure 4A). The population that displayed lower SSC from the rest of the CD162+ virus population was not included in the gating strategy, since it was expected that these events were EVs. Importantly, we quantified the fluorescence of each virus in units of median PE MESF and presented them with robust standard deviation (SD), since this SD calculation is not as skewed as the conventional SD due to outlying events that could be caused by background or coincidence. The PE MESF values ± robust SD for each virus were as follows: α4β7+ viruses with 40 ± 38.5 MESF, CD14+ HIV with 20 ± 12.1 MESF, and CD162+ HIV with 100 ± 57.5 MESF (Figure 4B). MESF values for viruses with incorporated host proteins (Figure 4B, colored histograms) were derived from the events in the upper gates on the positively stained population of viruses (Figure 4A, bottom row). No substantial fluorescence shift was detectable on our control virus and the majority of labeled control virus events fell within the lower fluorescence gate. Since 99% of our unstained virus events also fall within this gate, low levels of labeling that are present within this lower PE MESF gate are likely below the range of detection for this instrument and will appear as background fluorescence. Due to this, we concluded that the MESF values for our control viruses were in the range of background fluorescence (<10 MESF) for our instrument. Assuming that one antibody has one molecule of PE associated with it (fluorophore-to-protein ratio of 1; F/P) [67], the PE MESF values associated with each virus are representative of the number of antibodies bound to each virus, providing a close estimation of the number of host proteins present on each virion, assuming no steric hindrance [73,74]. Notably, the three different viruses containing host proteins (α4β7+, CD14+, and CD162+) all displayed different levels of median PE fluorescence, with CD162+ viruses displaying the highest levels of staining (Figure 4A,B). This finding was in line with the trend that was shown in our immunomagnetic capture assay, suggesting that the data generated in our flow virometry staining are representative of a well-established technique for phenotyping virions. However, the MESF of the CD14+ virus was found to be lower than that of α4β7+ HIV, which was different from the results seen in the virus capture assay. As an additional measure to ensure that the populations seen in our flow virometry plots were not the result of coincidental events, viral aggregates, or particle swarming [45], we diluted our samples 1000-fold (5 × 10^5^ particles/mL) after staining and before sample acquisition (Figure 4AD), and the population of each stained virus remained consistent in terms of fluorescence and distribution across multiple dilutions. This suggests that viral swarming and aggregates were not present at notable levels in the sample labeling. Undiluted stained samples with higher particle concentrations were not shown here since higher levels of coincidental events are seen when samples that are overly saturated with particles and antibodies are run on the cytometer.

As noted, single staining of cellular proteins on HIV virions revealed distinct phenotypes of the virus populations containing the host proteins (Figure 4A, bottom row), indicating that there may be some unique features in the biology of these virus populations. To confirm that the acquired data were representative of our virus samples and not contaminating vesicles, we decided to assess the presence of EVs in our virus preparations. To investigate this while simultaneously ascertaining the ability of our flow virometry methods and detection system to interrogate two antigens at the same time on individual virions, we chose the CD81 tetraspanin as a secondary antigen to identify the presence of EVs in our samples. CD81 was selected since it is endogenously expressed on our virus producer cells and has been previously used in numerous studies as an EV marker [51,75,76,77,78]. Furthermore, we did not observe any appreciable levels of virus capture when anti-CD81 was employed in our antibody-mediated virion capture assay, indicating the high likelihood that CD81 is not present on pseudovirions produced in HEK293 cells (Appendix A). To begin, we verified our ability to detect non-virus EV populations with the anti-CD81 antibody using supernatants collected from cells transfected with an empty vector (Figure 4C, mock-transfected), which produced events that overlapped in side scattering profiles with our virus population (Figure 4C, control HIV). Since the viral construct SG3^Δenv^ was not included in the mock transfection, any positive labeling in mock-transfected samples can be attributed to EVs. Labeling these samples with a BV421-conjugated anti-CD81 antibody through single staining showed some labeling for CD81 (Figure 4C upper gates), which confirmed the presence of CD81-positive EVs contaminating our virus samples. In an effort to distinguish contaminating EVs from our virus events when double staining, we performed a gating strategy (outlined in Appendix A) that distinguished total virus populations based on light scattering and PE fluorescence, and then compared levels of CD81 and transfected host proteins on the virus and EVs through single and double staining. Since mock-transfected cells were shown to produce CD81+ EVs (Figure 4C), we phenotyped the surfaces of these EVs, which are present in the virus gate due to their shared similarities in light scattering properties with the transfected viruses. Low levels of CD81 staining were seen on mock-transfected supernatants, but none of the host proteins of interest were detected, as expected (Appendix A).

In Figure 4D, we performed double staining of transfected cellular proteins and endogenous CD81. As expected, CD81 was not detected in abundance in the majority of virus events labeled for α4β7 and CD162 (few double positives in α4β7+ and CD162+ viruses), while in the case of CD14+ viruses, there was an unexpectedly large proportion of viruses that were double positive (CD14+ CD81+). We also observed different levels of CD81 staining on virus populations that were single-stained with anti-CD81, with CD14+ viruses containing more CD81 than α4β7+ and CD162+ viruses (Appendix A). It is possible that the differences in CD81 labeling may be due to biological differences in the viruses or distinct molecular mechanisms of incorporation, questions which are beyond the scope of this manuscript. Taken together, our double staining results demonstrate that we can effectively gate virus populations and stain multiple cellular proteins on individual virions using virus samples collected as culture supernatants, without the need for any additional processing or labeling of virus samples (nucleic acid or membrane stains) in order to analyze the discrete virus population.

## 4. Discussion

The novel methods described in this study enable virus phenotyping with a simple overnight staining protocol and flow cytometric detection, providing the most precise characterization and quantification of proteins on individual HIV virions to date. Here, we show that flow virometry is a tool that can provide sensitive analysis of single HIV particles to supplement current techniques for studying viruses. Furthermore, while most studies of cellular proteins in the HIV envelope are semi-quantitative [15,17,18,27], here we show that when calibrated with fluorescence quantitation beads, a measure of the number of antibody molecules bound to virus-associated proteins can be approximated, enabling the quantitation of proteins on individual virions.

Although staining of cellular proteins on HIV has been performed in the past [41,52,53], the protocol used in this study did not require ultracentrifugation to concentrate the virus or the use of MNPs, as in previous studies. While an ultracentrifugation protocol may serve as a barrier to routine laboratory operations, it may also alter infectivity by inducing gp120 shedding, which may be especially important in studies that aim to sort infectious virions after flow virometry analyses [79,80,81]. Similarly, a flow virometry protocol that eliminates centrifugation would be beneficial, since certain viruses generate viral aggregates during centrifugation, which could negatively impact the acquired data [82]. Detection of viruses without the need for MNPs allows for more sensitive quantitation, since all virions can be assayed without the need to capture a virus with a primary antibody before performing sample analysis with a second antibody. Despite the sensitivity of our protocol on the cytometer, proteins that are present at very low levels on the viral surface may still be below the level of detection for staining, regardless of the flow virometry labeling protocol used. Since the instrument’s background fluorescence generated from running cell culture media ends at ~10 PE MESF, we will likely not be able to detect anything with less than ~10 molecules on the surface, since it would not be distinguishable from background. Importantly, because the units that we report herein are standardized due to calibration, research groups at other institutions can perform similar experiments and compare their results directly to ours, irrespective of whether their instrument is more or less sensitive than ours.

As this study reports an antibody-based method for labeling virion-incorporated proteins, a few limitations and considerations revolving around antibody interactions should be noted. Since the quantitation of host proteins in this study uses bivalent antibodies, MESF data reported herein may be off by a factor of two. Similarly, the antibody clone used for labeling can lead to divergent results. This is especially important to consider in future flow virometry studies of HIV-1 gp120, which exhibits variable levels of antibody binding, depending on the conformational state of gp120 and the anti-gp120 antibody clone used [83,84]. These considerations are particularly important for nanoscale particles that have relatively low numbers of proteins on their surface (due to size limitations), whereby any small variance in antibody labeling can drastically skew the final result. Most importantly, while reports in units of MESF offer an informative estimation of the number of antibodies bound to the surface of a virus, this may not always be equal to the absolute number of proteins in the viral envelope [74]. This may be most relevant for proteins such as CD162, which have large extracellular domains (~50 nm) that may allow for better antibody accessibility or for the binding of multiple antibodies [85,86]. Additionally, while we assumed that the fluorophore-to-protein (F/P) ratio of the antibodies used in this study was 1, for applications that necessitate extremely precise levels of quantitation, the antibodies employed should be assayed by spectrophotometry to validate the F/P ratio.

Another major consideration for this technique is the presence of EVs in viral preparations. Here, we attempted to use the CD81 tetraspanin as a marker to discriminate between viruses and EVs. Staining of CD81 allowed for the identification of CD81-positive events, which we assumed would be representative of EVs. However, unexpectedly, we observed double-positive populations (CD81+ and host protein+), which seemingly indicated that either EVs express our cellular proteins of interest or that a fraction of our viruses contain CD81 within their envelope. While CD81 is a commonly used EV discrimination marker [51,75,76,77,78], it has also been reported to be present in T-cell-derived viral preparations [87], which may explain the low levels of CD81 staining observed in our virus population. However, with virion capture assays, anti-CD81-antibody-based capture did not yield appreciable level of virus capture, despite the same virus samples staining positive for CD81 in flow virometry. Differences in the results seen from these two techniques may be due in part to differences in epitope accessibility when viruses are bound to magnetic beads (as used in the capture assay) versus when they are free in suspension (flow virometry), and this disconnect remains the subject of ongoing investigations. The CD63 tetraspanin was also tested as an EV marker in this study, but no staining on our virus preparations was detectable. While CD81 is commonly used as an EV marker [51,75,76,77,78], the similar biogenesis pathways that retroviruses and EVs share can result in common cellular markers on both types of small particles [88]. Our work demonstrates that CD81 is not a marker that is exclusive to EVs, which is an important consideration when discriminating between viruses and EVs in future flow virometry studies. Notably, while it could not be used here since our viruses were not produced in primary CD4+ T cells, the leukocyte antigen CD45 has been used successfully in the past in flow virometry assays as an EV discrimination marker in PBMC-produced viruses, while CD45 has reliably been shown to be excluded from HIV virions in other assays as well [27,41,89,90]. Thus, in primary virions, CD45 may serve as a better EV discrimination marker than CD81, since CD81 appears to be present on a lower number of virions, as demonstrated by the flow virometry results in this manuscript. While we were able to perform fluorescence calibration with rainbow calibration particles to report BV421 fluorescence in standardized units of ERF rather than MESF, to be able to compare relative levels of CD81 on our virus particles to our host proteins of interest would require the comparison of single stains using the same fluorophore (i.e., all antibodies conjugated with PE). However, as our goal was to identify EVs using CD81 staining and not to enumerate the CD81 proteins on EVs, this experimental question was beyond the scope of this technical manuscript.

Finally, while the viruses here displayed substantial levels of incorporated host proteins, it is possible that clinical isolates of viruses may display very different levels of incorporated host proteins, particularly when they are derived from virus producer cells that are not in the context of overexpression (from transfection). Our ongoing work is focused on the detection of incorporated proteins on primary HIV-1 isolates, including detection of viral gp120, in order to determine the relative ratios of host and viral proteins on viral surfaces. Given that trends observed in our virion capture assays were reproducible in terms of flow virometry, and that flow virometry appeared to be a more sensitive assay to detect cellular proteins incorporated at low levels (e.g., CD81), we anticipate that the incorporation of cellular proteins on primary viruses should be readily detectable with flow virometry, particularly when those proteins have been previously validated in virus capture assays. Numerous studies have employed virion capture assays to phenotype the surfaces of primary viruses, indicating that primary virions incorporate a broad range of cellular antigens to detectable levels that could be quantified by emerging flow virometry methods. We anticipate that this technical advance herein will open the field for phenotyping viral surfaces with flow virometry, offering a relatively rapid, robust, and quantitative method, with several advantages for high-throughput experimentation.

## Figures and Tables

**Figure 1 viruses-12-01296-f001:**
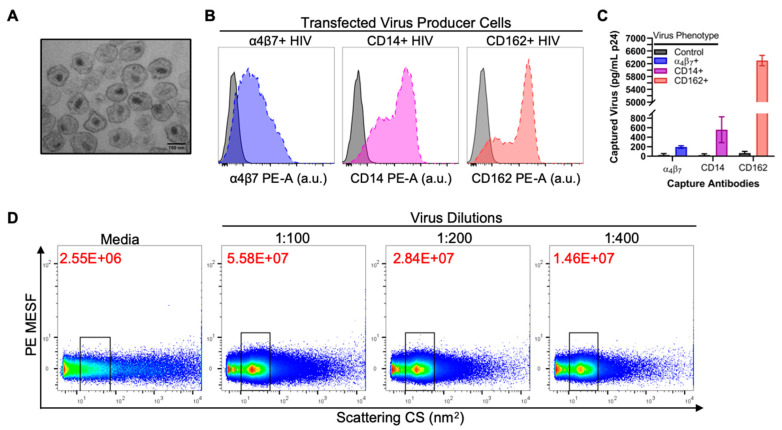
Validation and quality control of viral stocks to be used in flow virometry. (**A**) HIV-1 pseudoviruses (SG3^Δenv^) used in this study as visualized by transmission electron microscopy. Image shown represents free virions from one section of virus-producing HEK293 cells and was reproducible across 5 different cell sections that were imaged. (**B**) Cell surface expression of transfected host proteins (α4β7, CD14, or CD162) as analyzed by flow cytometry. Colored dotted line histograms indicate positive staining when the respective host protein was co-transfected (α4β7+, CD14+, or CD162+) with SG3^Δenv^, while control (grey solid line) histograms indicate negative staining with the anti-host protein antibodies on cells producing control virus (SG3^Δenv^ alone), without co-transfection of host protein. (**C**) Detection of virion-incorporated host proteins by antibody capture assay, with each virus type indicated by a different bar color (“virus phenotype” containing different host proteins). Control virus contains no host protein, only the SG3^Δenv^ backbone. Captured virus is presented as the amount of p24 (pg/mL) after lysis of bead-associated virus. Data shown indicate the mean level of virus capture +/− SD for duplicate samples after removal of the background noise (non-specific IgG capture). (**D**) Serially diluted (1:100, 1:200, 1:400) preparations of unstained control HIV are shown with gating on the pseudovirus population. Particle concentrations (particles/mL) were calculated based on events in the gated regions when acquired at a sample rate of 10 µL/min for 1 min, denoted in red on each dot plot. Media alone is shown to indicate the level of instrument noise and background noise present.

**Figure 2 viruses-12-01296-f002:**
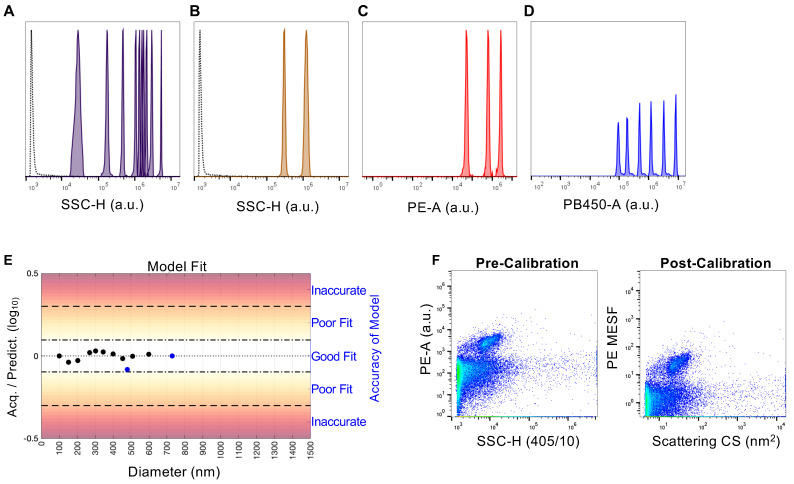
Standardization of flow virometry data with reference beads for light scatter and fluorescence calibration using FCM_PASS_ software. (**A**) NIST-traceable polystyrene (100, 152, 203, 269, 303, 345, 401, 453, 508, 600 nm) and (**B**) silica calibration beads (480, 730 nm) used to perform side scatter calibration. Bead populations are displayed from left to right in order of increasing diameter (i.e., higher arbitrary unit, a.u.). The empty dotted histogram denotes instrument background noise from PBS. (**C**) Quantibrite PE beads and (**D**) Rainbow calibration particles used for fluorescence calibration, as detected with the same gains used for virus sample acquisition on the flow cytometer. Bead populations are listed from left to right in order of increasing fluorescence (higher a.u.). (**E**) FCM_PASS_ output assessing the fit of our data generated on the CytoFLEX cytometer for scatter modeling by comparing the predicted versus acquired scatter values for the sizing beads of differing diameters (black and blue data points represent polystyrene and silica beads, respectively). (**F**) Representative plots comparing uncalibrated data expressed in arbitrary units of fluorescence intensity to calibrated data in standard units (MESF, nm^2^), using CD14+ HIV stained with an anti-CD14-PE antibody as representative data. Fluorescence and light scattering calibrations were performed on all datasets used in this manuscript, with consistent calibration values generated across all datasets.

**Figure 3 viruses-12-01296-f003:**
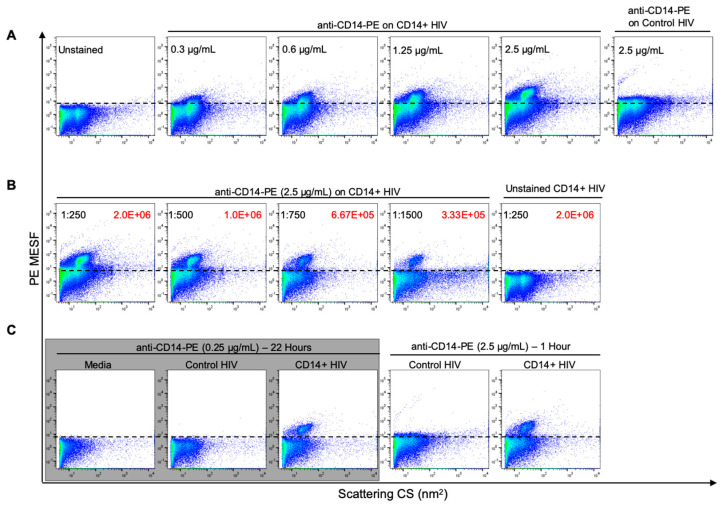
Optimization of antibody labeling protocol for phenotypic analysis of HIV pseudoviruses by flow virometry. (**A**) Titration of an anti-CD14-PE antibody on CD14+ HIV, stained for 1 h at room temperature (RT). Unstained virus and a control virus (without CD14) stained at the maximum concentration (2.5 µg/mL) of antibody tested are shown for comparison. (**B**) Dilution series of CD14+ virus labeled with 2.5 µg/mL of anti-CD14-PE to demonstrate reduction of background fluorescence from coincidence. Virus dilutions are shown in black, with associated particle concentrations (particles/mL) shown in red. (**C**) Reduction of coincidence through ten-fold reduction in antibody concentration and increased staining time, as seen when comparing the data in the grey box (left three plots) to the right panel (two plots). The staining time was increased from 1 h at RT to overnight at 4 °C to obtain an equivalent level of labeling as seen in (**B**). This optimized protocol (denoted with the grey box) was used for all subsequent staining procedures. Events above the dashed lines indicate positive labeling or increased background fluorescence due to coincidence. This line was set directly above the level of background fluorescence seen on unstained virus or cell culture media. For each panel, a range of dilutions were tested to ensure that the observed results were reproducible across multiple conditions and were not due to coincidence.

**Figure 4 viruses-12-01296-f004:**
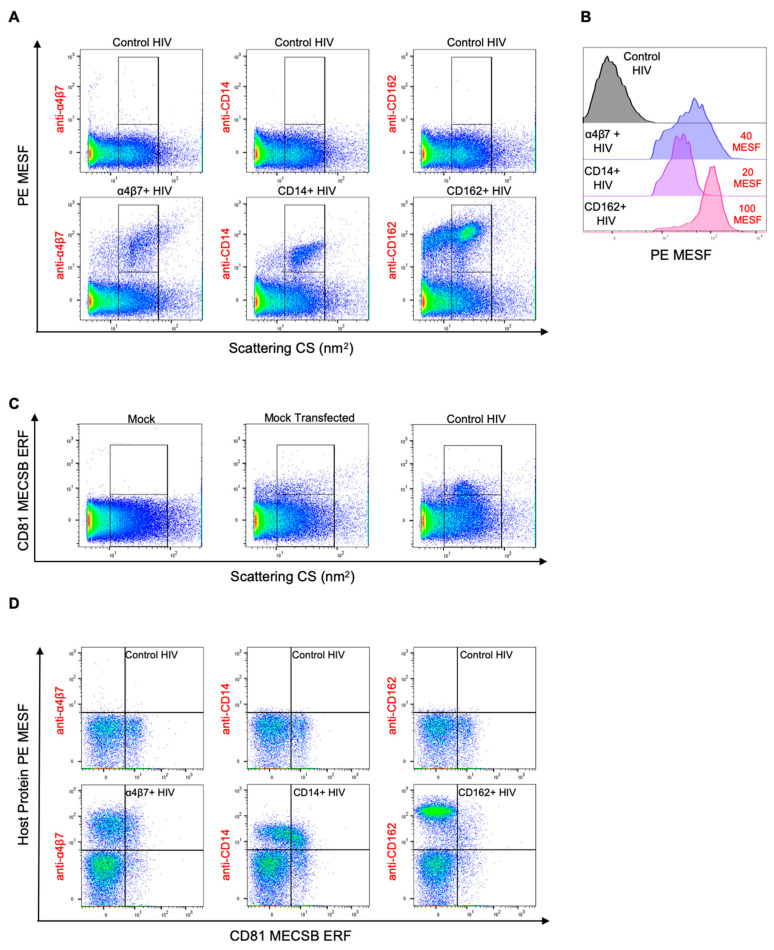
Phenotypic analysis of cellular proteins on the surface of HIV pseudoviruses. (**A**) Single staining of cellular proteins on virions with incorporated host proteins (α4β7+ HIV, CD14+ HIV, CD162+ HIV) using PE-conjugated antibodies specific for the respective incorporated cellular proteins. Lower gates are set for the side scattering population of control viruses. The lower gate spans 10–60 nm^2^ on the x-axis and has an upper limit of 10 PE MESF on the y-axis. Upper gates display positive host protein staining as determined using the control virus (in top row) and extend above the background fluorescence (>10 PE MESF on the y-axis). (**B**) A comparison of cellular protein levels on each of the transfected virus populations, as identified from the upper gates in (**A**), with the median PE MESF value of each population shown in red (top right). The control virus (grey) was identified using the lower gate from (**A**). (**C**) CD81 tetraspanin staining of cell culture media, mock-transfected cell supernatants, and control HIV. Transfection of HEK293 cells with a mock vector induced the release of CD81-positive non-virus particles, as identified by the upper gate. The lower gate was set on events generated by acquiring cell culture media alone. (**D**) Dual labeling for transfected cellular proteins and tetraspanin in HIV viruses using PE-conjugated antibodies against cellular proteins (α4β7, CD14, CD162) and a BV421-conjugated anti-CD81 antibody. BV421 is expressed in equivalent reference fluorophore (ERF) units of molecules of equivalent cascade blue (MECSB). The events shown are representative of the total virus populations from both gates in (**A**). Gating controls are shown in Appendix A. The results are representative of three technical replicates.

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
