# Peer review of "Flow Virometry Quantification of Host Proteins on the Surface of HIV-1 Pseudovirus Particles"

_viruses, 2020, doi:10.3390/v12111296_

Round 1

Reviewer 1 Report

In this work, Burnie and colleagues optimized a new protocol to detect viral particles using flow virometry. This protocol allows the detection of the viruses directly in the supernatants of the producer cells, without the need for other purification methods. This is undoubtedly a great improvement in this technique.

My major concern is the sensitivity of this protocol. In this work, they only used pseudo-particles produced by HEK293 cells co-transfected with the proviral genome and the cellular protein of interest. All the cellular proteins were highly expressed by the producer cells and possibly they were incorporated in viral particles at a higher concentration than in physiological conditions. In support of this concern, when they analyzed the incorporation of CD81, endogenously expressed by 293HEK cells, the sensitivity of the staining appeared much lower, and the detection of the positive population was much less net.
Authors should show how they detect cellular proteins in the viral population produced by the natural target of the virus, such as T cells (they could use any T cell line, or ideally primary CD4+ T cells). Or, at least, titrate the amount of co-transfected cellular proteins and evaluate if this affects the detection into viruses. In this case, co-transfecting a membrane cellular protein that is not incorporated into viral particles should be also included as a negative control.
Minor points :
- Is any particular reason why the authors used 293HEK cells to produce the viral particles, instead of the 293T-HEK cells? If yes, this should be explained in the manuscript.
- For TEM experiments, it is stated that " HEK293 cells were grown on sterilized 18 mm diameter coverslips (…) and 24 hours after transfection media was removed from the coverslips and the cells were washed with PBS ". Are the viruses shown in figure 1A cell-associated? Why they did not analyzed viruses in the supernatants?
- In figure 3B, it would be more informative to indicate the number of detected events instead of the concentration of viral particles used for the staining.
- In figure 3C it seems that the number of particles detected with the optimized protocol is lower than when using the 1h protocol. Indicating the number of detected events would help to compare the two methods.
- The number of replicates for each experiment should be more clearly indicated.

Reviewer 2 Report

This is an interesting report by Burnie et  al  that describes  detection of host proteins on  pseudovirus particles.

  1. Have some questions for the results with 22h staining, since to establish incubation time that long you have to know kd and ka for the ab-ag. During that time ab can bind and dissociate from ag many times. So basically not clear how you selected  22h and not 10h or 15h. should provide some kinetics for that.
  2. cell surface and intracellular staining for cd81 should be shown of HEK293.
  3. Cd81 is not good selection as an ev marker since it is known that hiv virions do incorporate cd81.
  4. Overall not sure that these particles can be called viruses, since they lack gp120. While with gp120 cellular markers may incorporate differently in bilayer that is taken up by virus.
  5. Line 340 do not see any evidence/prove that “populations appeared biologically distinct”
  6. Double staining experiments are not convincing.

a.Line 401 I believe figure S5B has no staining with cd81

b.Not sure how you explain the single stain a4b7 25% vs 20% double stain, more striking for CD162 54.4% single stain vs 34.4% double stain and disappearing CD81 from 20 to ~8%, and from18% to ~6%

c.No statistics is shown, avg +/- for performed experiments should be included

d.Figure 4D and supplemental fig 5(D)showing same data

Round 2

Reviewer 1 Report

In the revised version of the manuscript, the authors answered my previous concerns.